# Investigation of Gastrointestinal Toxicities Associated with Concurrent Abdominal Radiation Therapy and the Tyrosine Kinase Inhibitor Sunitinib in a Mouse Model

**DOI:** 10.3390/ijms25031838

**Published:** 2024-02-02

**Authors:** Amber R. Prebble, Bailey Latka, Braden Burdekin, Del Leary, Mac Harris, Daniel Regan, Mary-Keara Boss

**Affiliations:** 1Veterinary Teaching Hospital, Colorado State University, Fort Collins, CO 80523, USA; amber.prebble@colostate.edu; 2Department of Clinical Sciences, Colorado State University, Fort Collins, CO 80523, USAbraden.burdekin@colostate.edu (B.B.); 3Department of Environmental and Radiological Health Sciences, Colorado State University, Fort Collins, CO 80523, USA; del.leary@colostate.edu; 4Department of Microbiology, Immunology, and Pathology, Colorado State University, Fort Collins, CO 80523, USA; mac.harris@colostate.edu (M.H.); daniel.regan@colostate.edu (D.R.)

**Keywords:** radiotherapy, crypts, intestine, proliferation, vascular density

## Abstract

Tyrosine kinase inhibitors (TKIs) may be combined with radiation therapy (RT) to enhance tumor control; however, increased incidences of gastrointestinal (GI) toxicity have been reported with this combination. We hypothesize that toxicity is due to compromised intestinal healing caused by inhibition of vascular repair and proliferation pathways. This study explores underlying tissue toxicity associated with abdominal RT and concurrent sunitinib in a mouse model. Four groups of CD-1 mice were treated with 12 Gy abdominal RT, oral sunitinib, abdominal RT + sunitinib, or sham treatment. Mice received oral sunitinib or the vehicle via gavage for 14 days. On day 7, mice were irradiated with 12 Gy abdominal RT or sham treated. Mice were euthanized on day 14 and intestinal tract was harvested for semiquantitative histopathologic evaluation and immunohistochemical quantification of proliferation (Ki67) and vascular density (CD31). Non-irradiated groups had stable weights while abdominal irradiation resulted in weight loss, with mice receiving RT + SUN having greater weight loss than mice receiving RT alone. Semiquantitative analysis showed significant increases in inflammation in irradiated groups. The difference in the density of CD31+ cells was significantly increased in RT alone compared to SUN alone. Ki67+ density was not significant. In summary, we identify a lack of angiogenic response in irradiated GI tissues when abdominal RT is combined with a TKI, which may correlate with clinical toxicities seen in canine and human patients receiving combined treatment.

## 1. Introduction

Radiation therapy (RT) is a mainstay of cancer treatment. In the process of targeting and killing cancer cells, RT can lead to normal tissue damage, the severity of which depends on dose, fractionation, and the organs at risk in the treatment field [1]. Abdominal RT can lead to gastrointestinal (GI) mucositis, which can cause adverse GI side effects such as nausea, vomiting, and diarrhea [2]. Radiation-induced damage to epithelial stem cells located in the base of intestinal crypts can result in clonogenic cell death due to apoptosis of crypt epithelial cells [3]. RT can also induce apoptosis of irradiated GI microvascular endothelial cells, which release cytokines, chemokines, and growth factors as they apoptose [4,5]. Additionally, RT administered to the GI tract causes inflammation as local immune cells infiltrate the area and release cytokines and chemokines as part of the inflammatory response [2,4,5,6,7].

Another method of treating cancer that emerged over two decades ago is through the inhibition of naturally occurring tyrosine kinases. Tyrosine kinases are enzymes which are often overexpressed in cancers and activate many proteins [8,9]. This activation occurs when a phosphate group is transferred from ATP to the tyrosine group on the receptor tyrosine kinase (RTK) [8]. RTKs span the cell membrane with extracellular and intracellular domains connected by a hydrophobic transmembrane domain [8]. RTK activation induces signaling of pathways associated with proliferation, such as Ras, Raf, MEK, mTOR, and MAPK, as well as angiogenesis pathways such as those associated with vascular endothelial growth factor (VEGF) [10,11,12]. These cellular proliferation and angiogenesis pathways play an important role in wound healing in normal tissues, but can lead to tumor growth and metastasis in tumor microenvironments [9,13,14]. Tyrosine kinase inhibitors (TKIs) compete with the ATP, preventing phosphorylation, and act as an off switch to prevent proliferation and growth factor signaling [8]. The TKI sunitinib specifically targets the endothelial growth factors VEGFR1, VEGFR2, and VEGFR3, platelet-derived growth factors (PDGF) PDGFRa and PDGFRb, as well as KIT, FLT3, and CSF1R [11,15,16,17].

In cancer therapy, TKIs can be used as a single agent or in conjunction with RT to enhance tumor control, both locally and systemically, due to anticancer and antiangiogenic effects [10,14,18]. TKIs are an appealing cancer treatment because formulations can be chosen to target a particular pathway of interest, allowing for selective inhibition of proteins found specifically in a certain cancer; or a more broadly targeted formulation may be chosen for selective inhibition of many pathways, potentially decreasing risk of developing resistance [14]. When antiangiogenic TKIs were first developed, it was hoped that they could improve the efficacy of RT due to normalization of vasculature in solid tumors [14,19]. Tumor types explored for such treatments included rectal carcinoma, glioblastoma, hepatocellular carcinoma, and lung cancers [20,21]. These early concurrent treatment protocols used conventionally fractionated RT, and more recent protocols have utilized stereotactic body radiation therapy (SBRT) and whole brain radiation therapy (WBRT) [12,19,21,22,23]. When TKIs are administered in combination with RT, favorable tumor response and progression-free survival times have been reported in human and veterinary clinical trials [24,25,26,27,28,29]. Tumors treated in these studies in which the investigational therapeutic protocol involved some combination of TKI and RT have included intracranial metastatic non-small cell lung cancer (NSCLC) [24], primary NSCLC [26], metastatic renal cell carcinoma [28], and head and neck squamous cell carcinoma [29] in human cancer patients, and inflammatory mammary carcinoma [25], cutaneous mast cell tumors [27], and nasal carcinoma [30] in canine cancer patients. More recent publications support the use of concurrent fractionated RT, WBRT, or stereotactic radiosurgery with TKI to treat primary and metastatic NSCLC [31,32]. While the previously mentioned studies conclude that this combination is effective with respect to tumor response and survival times, there have also been reports raising concern for unanticipated toxicities. Pneumonitis has been reported in human medicine when thoracic RT was combined with EGFR-TKIs [26,31]. Significantly increased rates of radiation necrosis have been seen in treatment of brain metastases with SRS when combined with VEGF TKIs or EGFR TKIs [33]. Increased GI toxicities, including bowel perforation, have been seen with SBRT involving the abdomen when administered in combination with TKIs [1,12,22,34]. Notably, a phase I study investigating the combination of the TKI sorafenib with SBRT for treatment of hepatocellular carcinoma concluded that this treatment combination should not be used concurrently due to significant toxicities, including death [21]. In veterinary medicine, the TKI toceranib is commonly used to treat mast cell tumors (MCT) and is also used off-label for treatment of various other tumor types including sarcomas and carcinomas, sometimes in combination with RT [25,27,35,36,37]. At our institution, we suspected an increased incidence of GI toxicity when toceranib was administered to canine patients that were receiving concurrent abdominal RT. Upon retrospective review of canine cancer cases, we found significant increases in rates of diarrhea, vomiting, and hyporexia with this combination when compared to dogs treated with toceranib alone, abdominal RT alone, or RT administered to sites outside of the abdomen [38]. Although the GI toxicity associated with abdominal RT and concurrent TKI administration has now been documented in human and canine cancer patients, the underlying biological and pathological processes of this toxicity remain unknown.

Both TKIs and RT affect the vasculature of normal tissues and alter endothelial cell proliferation, which impacts angiogenesis and wound healing [3,10,39,40,41]. Abdominal radiation causes vascular permeability within 24 h, with this acute phase predominated by endothelial apoptosis [42,43,44]. After irradiation, an inflammatory response is mounted and endothelial proliferation is initiated, leading to a vascular response and the creation of new capillaries that are irregular in shape and diameter [41]. An increase in VEGF expression, an indicator of angiogenesis, has been shown in rectal biopsies taken 1–3 days following completion of 72–74 Gy total dose fractionated 3D conformal RT in human prostate cancer patients [45]. With respect to TKIs, the lining of the GI tract is sensitive to TKI inhibition of endothelial growth factors, such as those in the VEGF family, due to their direct impact on endothelial cells and indirect impact on epithelial stem cells in intestinal crypts [46]. TKIs can inhibit angiogenesis [10], which can lead to increased healing times due to lack of vascular formation in response to wound development [12,39,47]. Signaling pathways that lead to proliferation, such as Ras, Raf, MEK, mTOR, and MAPK, targeted by TKIs, play an important role in the resolution of inflammation, re-epithelialization, and healing [13]. Therefore, we hypothesized that abdominal RT in combination with the TKI sunitinib would lead to increased GI toxicities due to compromised intestinal healing caused by inhibition of both vascular repair and proliferation pathways in the irradiated normal GI tissues. In this study, we explore the normal tissue changes underlying gastrointestinal toxicity associated with abdominal RT and concurrent treatment with the tyrosine kinase inhibitor, sunitinib, in a preclinical mouse model.

## 2. Results

### 2.1. Defining Abdominal RT Dose

The purpose of this step was to identify an optimal abdominal RT dose for experimentation in CD-1 mice. Our aim was to ascertain a dose that led to mild weight loss, from which mice were recovering by day 7, as well as mild GI inflammation and mild crypt loss seen histologically. The percent weight change seen for each mouse from day 0 to day 7 post-RT are as follows: 0 Gy (103%, 104%), 4 Gy (107%, 104%), 6 Gy (105%, 102%), and 8 Gy (102%, 102%) (Figure 1a). We saw that mice exposed to 0, 4, 6, or 8 Gy abdominal RT had stable weights and no histological changes were seen in any part of the intestinal tract at the time euthanasia. Mice that were exposed to 11 Gy abdominal RT maintained stable weights (95%, 100%) and mild histologic changes were seen in the LI. Those that were exposed to 13 Gy had moderate weight loss (88%, 86%) and moderate histologic changes in the LI, which included blunted villi, moderate inflammation, and decreased crypt mucosal architecture. Mice that received either 14 or 15 Gy had severe weight loss (80%, 83%) and were euthanized on day 5 due to unacceptable weight loss and morbidity scores. Histology revealed severe GI tissue destruction, which were progressive changes from what was noted in tissues exposed to 11 and 13 Gy (Figure 1b). Based on the evidence of very mild weight loss and mild histologic changes seen at 11 Gy and more moderate weight loss and histologic changes at 13 Gy, we selected 12 Gy for the RT arms of this study to test our hypothesis that adding sunitinib would lead to increased toxicities.

### 2.2. Radiation Therapy + Sunitinib

#### 2.2.1. Weights

The mean weight and 95% CI for each group over the 14-day protocol is as follows: control 32.8 g (28.9–36.6 g), SUN alone 32.8 g (28.8–36.8 g), RT alone 33.9 g (28.6–39.3 g), and RT + SUN 31.6 g (28.0–35.2 g). The mean percent weight change among mice in control or sunitinib alone groups were stable (97% and 100%, respectively). Both groups that received abdominal RT lost weight, with those receiving RT + SUN losing more (88%) than those receiving RT alone (92%). There was a statistical difference in weight loss across the treatment groups (*p* = 0.0001), and upon multiple comparisons, statistically significant differences were seen, as the control group lost more weight than the SUN group (*p* < 0.0001), the RT group lost more weight than SUN (*p* < 0.0001), RT + SUN lost more weight than SUN (*p* < 0.0001), and RT + SUN lost more weight than RT (*p* = 0.0258) (Figure 2).

#### 2.2.2. Quantitative Histology

SI villi lengths and colon crypt area were measured as an assessment of repair response after RT, which has been shown to be impacted by microvascular function and the loss of crypt cell proliferation [48,49]. When exposed to RT, SI villi are expected to be shortened between 36 h and 3.5 days after RT, as the crypts fail to replace older cells sloughing from the tips of the villi [50]. The difference in SI villi length across groups at day 14, seven days post-abdominal RT or sham RT, was not significant (*p* = 0.605) (Figure 3). The difference in colonic crypt area compared across the treatment groups was significant (*p* < 0.0001). No significant difference in crypt area was found between the two groups that did not receive RT (control, SUN) (*p* = 0.9997) and there was no significant difference between the two groups exposed to abdominal RT (RT, RT + SUN) (*p* = 0.9935). There was however significance with every comparison between a group not receiving RT with a group that did receive RT (control vs. RT, *p* = 0.0003; control vs. RT + SUN, *p* = 0.0003; SUN vs. RT, *p* = 0.0001; SUN vs. RT + SUN, *p* = 0.0001), as crypt area increased in the irradiated tissues (Figure 4).

#### 2.2.3. Semiquantitative Histopathologic Scoring of Gastrointestinal Toxicity

There was an absence of inflammation throughout the GI tract of mice in the control or SUN groups (all inflammation scores: 0), while mice in both groups receiving RT (RT, RT + SUN) demonstrated an increased presence of a mixed inflammatory response consisting of neutrophils, lymphocytes, plasma cells, macrophages, and giant cells to varying densities throughout all segments of the GI tract (inflammation score: RT (1–3), RT + SUN (1–3) (Appendix A). This inflammation score was significantly higher in groups receiving RT as compared to groups receiving sham irradiation (control vs. RT, *p* = 0.0043; control vs. RT + SUN, *p* = 0.0025; SUN vs. RT, *p* = 0.0006; SUN vs. RT + SUN, *p* = 0.0003). There was also a significant difference in crypt density between treatment groups (*p* < 0.0001), with marked crypt loss observed in all groups receiving radiation therapy, as compared to unirradiated controls (control vs. RT, *p* = 0.0043; control vs. RT + SUN, *p* = 0.0013; SUN vs. RT, *p* = 0.0006; SUN vs. RT + SUN, *p* < 0.0006). The presence of crypt ulceration was not significantly different across groups (*p* = 0.4552). The presence of crypt abscessation was significantly different (*p* = 0.0009), and when compared between groups, RT + SUN had significantly increased levels of abscessation when compared to groups not receiving RT (RT + SUN vs. control, *p* = 0.0076; vs. SUN, *p* = 0.0023). RT + SUN had more abscessation than RT alone, but did not reach significance (*p* = 0.0862). There was a statistically significant difference in the score of crypt hyperplasia/regeneration between groups (*p* = 0.0018). Control and SUN groups had scores of 0 and both groups receiving RT had scores of 0 to 3. There was significance only when groups not receiving RT (control, SUN) were compared to groups that did receive RT (control vs. RT, *p* = 0.0411; control vs. RT + SUN, *p* = 0.0223; SUN vs. RT, *p* = 0.0047; SUN vs. RT + SUN, *p* < 0.0023). Groups receiving RT had less hyperplasia/regeneration than those that did not. These results are summarized in Figure 5 and examples of histological changes seen in our semiquantitative scoring data are presented in Figure 6.

#### 2.2.4. Immunohistochemistry

The density of CD31+ cells within the GI tract was significantly different across groups (*p* = 0.0235). The RT alone group had an increase in CD31+ expression compared to the SUN group (*p* = 0.0252). The density of Ki67+ cells within the GI tract was not significantly different across groups (*p* = 0.1937) (Figure 7). However, on review of the tissues, it was observed that the Ki67+ cells were distributed differently throughout the GI tract when compared between the treatment groups. We used semiquantitative scoring to assess the presence of Ki67+ cells within the SI villi and LI crypts (Appendix A). In the SI, each sample was scored for the frequency Ki67+ cells were seen within the villi. (Figure 8a) When unblinded, the distribution of Ki67+ cells in the SI from the mice in treatment groups that did not receive RT were contained primarily in the crypts of the villi, whereas the mice in groups exposed to abdominal RT had Ki67+ cells dispersed throughout the length of the villi. The scoring showed significant difference across the groups (*p* < 0.0001). There was no difference in Ki67+ cells of the villi when non-irradiated groups were compared (control vs. SUN, *p* = 0.5581), when the two groups exposed to abdominal RT were compared (RT vs. RT + SUN, *p* = 0.4615), or for control vs. RT + SUN (*p* = 0.0530). However, there was significance seen in the following comparisons of a non-irradiated group to one exposed to abdominal RT, with RT increasing the incidence of Ki67+ cells in the SI villi (control vs. RT, *p* < 0.0022; SUN vs. RT, *p* = 0.0006; SUN vs. RT + SUN, *p* = 0.0035). The percentage of Ki67+ cells confined within the crypts of the SI was analyzed and significant difference was seen across the four groups (*p* = 0.0106) (Figure 9a,b). There was no difference in Ki67+ cells of the villi crypts when non-irradiated groups were compared (control vs. SUN, *p* = 0.9963) nor when the two groups exposed to abdominal RT were compared (RT vs. RT + SUN, *p* = 0.9964). However, a significant increase in Ki67+ cells was seen when RT + SUN was compared to SUN alone (*p* = 0.0296). In the LI, colonic crypts were scored as having Ki67+ cells confined to the periphery of the crypt closest to the basement membrane (score 0) or extending beyond halfway around the crypt (score 1) (Figure 8b). All crypts analyzed in non-irradiated groups scored a 0, with Ki67+ cells confined to the lower half of the crypt. In groups receiving abdominal RT, all crypts analyzed contained scored a 1, with Ki67+ cells present throughout the full circumference of small crypts and spanning the lower half to two-thirds of larger crypts. The overall differences were significant (*p* < 0.0001); however, this quantification showed no difference in location of Ki67+ cells between the two non-irradiated groups (*p* > 0.9999) nor between the two groups that received abdominal RT (*p* > 0.9999). Significant differences were seen when either group receiving abdominal RT was compared to either non-irradiated group (*p* < 0.0001 for all comparisons).

## 3. Discussion

In this study, we aimed to characterize the underlying normal tissue changes associated with GI toxicity when abdominal RT is combined with the TKI sunitinib in a mouse model. Our results identified significant differences between the group receiving RT + SUN and the SUN and RT groups in the area of weight loss, which could be interpreted as preclinical evidence of GI pathologic changes leading to morbidity. We identified a significant increase in crypt abscessation when RT + SUN was compared against the vehicle control or sunitinib alone; the increase in crypt abscessation approached significance when compared to RT alone. The increased crypt abscessation in mice treated with RT + SUN suggests a histopathologic effect of compromised healing with this treatment combination. We also investigated the microvascular density and cellular proliferation throughout the GI tract following treatment. Mice treated with abdominal RT developed significantly increased GI microvascular density following treatment, whereas this effect was not seen in mice treated with RT + SUN nor in mice in the control or SUN treatment groups. These RT + SUN-associated morbidity and GI pathologic effects have not been previously defined in preclinical animal models.

Differences in weight loss were statistically significant across the four treatment groups. Interestingly, the control group lost significantly more weight than the SUN group, and there was no difference in weight loss seen between control and the two RT groups. The mice in groups treated with abdominal RT lost the most weight. Both groups receiving abdominal RT lost significantly more weight than the SUN group; this abdominal RT-associated weight loss in mice treated with 12 Gy was expected due to the mild GI side effects documented in mice treated with 11–13 Gy in the dose–response experiment. However, the group of mice receiving concurrent abdominal RT + SUN lost significantly more weight than the RT alone group, which may align with the clinical toxicities seen in our prior retrospective study of canine cancer patients receiving concurrent abdominal RT and the TKI toceranib [38].

We saw no difference in SI villi lengths between all four of the treatment groups. The villi are expected to be blunted 3.5 days after RT due to normal tissue damage [50]. As such, our results were somewhat unexpected, but this lack of shortening could be due to the SI villi having fully repaired by day 7 post-RT, indicating the repair process was not hindered or slowed by the addition of sunitinib. Colonic crypt enlargement was identified in mice receiving abdominal RT when compared to non-irradiated mice. This enlargement of crypts after abdominal RT was not unexpected based on the work of Withers showing colonic crypts are regenerated and enlarged 5.5 days after irradiation [50]; however, the lack of significant difference between the two RT groups indicates that the addition of sunitinib did not exacerbate or hinder this normal response to RT.

Semiquantitative scoring of GI injury revealed differences when tissues from the two non-irradiated groups were compared to those receiving abdominal RT. Notably, treatment with SUN did not affect GI inflammatory scoring compared to control mice. Previously, in a mouse model of intestinal cancer, sunitinib administration at a dosing protocol of 30 mg/kg every other day for 9 weeks resulted in a reduction in various inflammation-related factors, such as IL-6, IL-1α/1β, TNFα, and IFN-γ at the mRNA level within the GI tract [51]. Alternatively, rats chronically exposed to sunitinib at a lower dose (1.5–15 mg/kg/day) for 13 weeks developed histological evidence of small intestinal injury characterized as glandular hyperplasia that was sometimes associated with inflammation of the intestinal wall; however, those exposed to 0.3–6 mg/kg/day for 6 months had no evidence of GI inflammation [52]. In our study, the inflammatory response did not show significant differences between RT and RT + SUN for any parameters but approached significance for the presence of crypt abscessation. Crypt abscessation occurs when inflammatory cells infiltrate the lamina propria surrounding the crypts [53] and has been documented in the normal rectal mucosa of human cancer patients two weeks after starting fractionated RT for non-gastrointestinal cancers with a radiation field which included the rectum [54]. Crypt abscessation was also identified in rats 5 days after single fraction 10 Gy abdominal RT [55]. Crypt abscessation has been seen in human cancer patients who report GI side effects while receiving TKI therapy [56]. It has been reported that radiation-induced crypt abscessation in rats correlates with clinical signs of diarrhea and weight loss [53,55]. In human cancer patients, histologic changes, including crypt abscess, have been associated with loose stools [54,57]. In our study, we used weight loss as a surrogate marker for GI clinical signs. Given the evidence of correlation between abscessation and clinical signs in rats and human cancer patients, we might consider the significant weight loss seen in mice receiving RT + SUN when compared to RT to be correlated with their increased rates of crypt abscessation.

There was a significant difference in the microvascular density of the GI tissues between SUN and RT groups. The mice exposed to abdominal RT had a significantly increased density of CD31+ cells compared to those treated with SUN. The results from the mice exposed to abdominal RT are consistent with an increased vascular response to RT. However, this vascular response was seen only in the mice receiving RT, not in mice treated with RT + SUN. This normal vascular response seems to have been suppressed by the presence of SUN in the RT + SUN group. Following GI irradiation, normal tissues undergo an inflammatory response which includes increased expression of CD31 [3]. The link between post-RT GI vascularization and inflammation has been made in previous studies based on increased numbers of CD31+ cells seen in rectal biopsies from human patients treated with RT for prostate cancer [45] and irradiated rectal tissues in mice [58]. Currently, there is a gap in knowledge of whether CD31+ expression is altered in normal GI tissues after administration of sunitinib or other TKIs and this is an area of potential future investigation. If the increase in vascularity seen in the group treated with RT alone was a response to abdominal irradiation, we would have expected to see increased vascularization in the GI tract in both groups receiving abdominal RT. The normal radiation response of tissues in the GI tract is an activation of vascular repair and angiogenic pathways to begin the processes of wound repair and healing; an absence of this response in the RT + SUN group could indicate that at least one of these pathways has been altered and may be associated with the increased GI side effects that have been reported following concurrent abdominal RT with TKI administration in human and veterinary medicine. Alternatively, it could be argued that the GI vascular response following abdominal RT is linked with radiation-induced GI toxicity, and, as there is a lack of angiogenic response in the RT + SUN group, perhaps sunitinib is suppressing the tissue irradiation damage and associated RT-induced GI toxicity. However, this is not supported by the results of this study which demonstrate significantly increased weight loss seen in the RT + SUN group and equivalent levels of GI inflammation seen in both groups of mice receiving abdominal RT. Further, when comparing the GI toxicities reported in our retrospective canine study, concurrent abdominal RT with TKI resulted in significantly increased GI toxicity compared to dogs treated with abdominal RT alone, not a protective effect [38].

Ki67 stains proliferating cells and is an indicator of regeneration [59,60,61]. Epithelial stem cells, located within the base of SI and LI crypts, divide asymmetrically and produce one daughter stem cell which remains in the crypt and one daughter cell that proliferates and differentiates at it moves up and out of the crypt [62]. The normal epithelial differentiation and migration of intestinal stem cells takes approximately 3–4 days [63]. The differentiating daughter cell moves higher up in the crypt and into the villus (SI), losing its stem cell characteristics as it progresses and matures, eventually reaching senescence and being sloughed off into the intestinal lumen [62,63]. Ki67+ cells are known to migrate within the crypts of both the SI and LI as a response to radiation [64]. The onset of epithelial regeneration is seen in the SI of mice 60 h after irradiation and at 3.5 days in the LI [50]. Otsuka and Suzuki showed a significant reduction in Ki67+ cells in the lower region of ileal and duodenal crypts in the SI at five timepoints, up to 72 h, after abdominal irradiation with 1 and 4 Gy and stable numbers of Ki67+ cells in the upper region at these doses. They did not see changes at a dose of 0.1 Gy. They noted no significant changes in the overall number of Ki67+ cells present in the crypts. In our study, the density of Ki67+ cells throughout the GI tract of treated mice was not significantly different between the groups. What was different, however, was the location of the Ki67+ cells themselves when non-irradiated groups were compared to those that received abdominal RT. In non-irradiated GI tissues, these cells were confined to the crypts of the villi in the SI and the lower half of the perimeter of colonic crypts. In irradiated tissues the Ki67+ cells had moved out of the villi crypts and into the villi themselves in the SI and in the LI had moved to greater than 50% of the periphery of the colon crypts. There was no significant difference in location between the two groups receiving RT, indicating that the addition of sunitinib did not affect this migration. In the SI, we examined the location of Ki67+ cells and graded them as confined to the crypts at the base of the villi or extending into the villi. Otsuka and Suzuki did not comment on Ki67+ cells within the villi; however, they did see the Ki67+ cells moving to the portion of the villi crypts closest to the villi with all doses. When the percentage of Ki67+ nuclei within the crypts of the SI was quantified, the groups receiving RT (RT alone or RT + SUN) had a higher percentage of Ki67+ cells compared to non-irradiated groups (control or SUN). The difference was significant only when RT + SUN was compared to SUN alone; however, the difference approached significance for RT alone compared to SUN alone; this observed trend of an increase in Ki67+ cells in groups receiving RT is in line with what would be expected as the crypts regenerate after irradiation [50]. The statistical relevance of the finding of increased Ki67+ cells in crypts from mice treated with RT + SUN vs. SUN may not be of importance when considering the overall trend of increased Ki67+ cells in the crypts of irradiated small intestine. Otsaka and Suzuki also documented similar results in the crypts of the colon over the same doses and time, with Ki67+ cells not occurring in the upper most area of the crypts of non-irradiated tissues, but having a significantly higher percentage of Ki67+ cells in this location at all timepoints up to 72 h for tissues receiving 4 Gy. We observed something similar for colonic crypts receiving 12 Gy harvested at 7 days after abdominal RT. The findings of Otsuka and Suzuki’s study are similar to our findings where we saw no significance in the percent-stained area between our four treatment groups, indicating stable numbers of proliferating cells, but did have significant differences in the location of the Ki67+ cells when non-irradiated groups were compared to those receiving abdominal RT. To our knowledge, there are no known studies reporting the location of Ki67+ cells in mouse GI tissues 7 days after RT, which we are reporting here.

This study has several limitations. Our results are limited by the single endpoint and timing of tissue harvesting post-treatment. Tissues harvested at earlier times after RT likely would have shown different states of vascular repair and proliferation. Alternatively, had we chosen a later time-point we may have seen more significance in weight loss between groups and/or differences in the time to weight loss recovery. We chose 7 days post-abdominal RT or sham irradiation as our endpoint in an effort to balance the desire for evaluating histological GI changes with biologic changes in weight. We also recognize that different RT fractionation and dosing could have given different results than our single dose of 12 Gy. We targeted the whole abdomen but recognize that a smaller field or targeted RT could have also produced different results. We identified two IHC markers to investigate; this could be expanded into other markers of angiogenesis and proliferation pathways such as VEGF and co-receptors, CD34, CD105, proliferating cell nuclear antigen (PCNA), Lgr5, and MCM2 [19,45,65,66]. Apoptotic markers such as TUNEL and p53 could also be considered [4,67]. This study is exploratory and descriptive at this point; future mechanistic studies could further define the drivers of the effects and toxicities seen.

Our study has translational potential as TKIs are commonly used in human medicine as a powerful treatment option for cancers with an overexpression of tyrosine kinases [8]. Concerns have been raised about the combination of TKIs and concurrent abdominal RT; however, this combination has also shown improved outcomes over either therapy used alone in primary and metastatic NSCLC, head and neck squamous cell carcinoma, and metastatic renal cell carcinoma in human cancer patients, and cutaneous mast cell tumors and inflammatory mammary carcinoma in canine cancer patients. In order to capitalize on this increased tumor control, it will be important to identify safe approaches to combining RT with TKIs. Further investigation is warranted into the clinical use of RT and TKIs to optimize timing of administration, dose of either modality, or fractionation of RT.

In conclusion, we report on increased weight loss and histopathologic changes throughout the GI associated with the combination of abdominal RT and the TKI sunitinib in a mouse model. We found a significant increase in weight loss in mice receiving SUN + RT when compared to mice receiving abdominal RT alone or sunitinib alone. We report a lack of GI vascular response following RT in mice receiving RT + SUN, indicating that the typical GI radiation-associated angiogenic response may have been hindered by the addition of sunitinib. We also quantified an increase in the incidence of crypt abscessation in mice treated with RT + SUN, and, while not statistically significant compared to RT, reveals a trend that suggests compromised healing of GI tissues in mice treated with RT + SUN. It is possible that clinical toxicities which have been reported with simultaneous abdominal RT and TKI administration could be attributed to an impaired tissue response and healing in irradiated normal tissues of the GI tract. Further study is recommended to determine whether approaches to minimize GI toxicity associated with abdominal RT and TKI treatment combinations are effective, while also preserving the beneficial biological therapeutic effects of concurrent RT and TKI.

## 4. Materials and Methods

### 4.1. Animal Husbandry

Six- to 14-week-old female and male CD-1 outbred mice (Envigo, Indianapolis, IN, USA) were used in this study. Mice were housed in accordance with animal welfare standards of the laboratory animal facilities at Colorado State University (CSU) (Fort Collins, CO, USA). Mice were provided with food and water ad libitum and exposed to a 12:12 h light–dark schedule. The experimental protocol was reviewed and approved by the CSU Institutional Animal Care and Use Committee.

### 4.2. Sunitinib (SUN) Preparation

The aqueous vehicle for sunitinib administration was a carboxymethylcellulose (CMC) formulation containing: 0.5% CMC, 1.8% sodium chloride, 0.4% Tween-80, and 0.9% benzyl alcohol dissolved in reverse osmosis deionized water. Sunitinib malate powder (LC Laboratories, Woburn, MA, USA) was added to the vehicle at concentration of 12 mg/mL and suspended via vortex. The vehicle and suspension were created weekly and stored in the dark at 4° C.

### 4.3. Irradiation

#### 4.3.1. Abdominal Irradiation

Six-week-old mice were randomized to treatment groups (n = 1–2/group). Mice were anesthetized with 2% isoflurane gas mixed with oxygen in an induction chamber and maintained via nosecone with the same. They were placed in an X-RAD SmART+ irradiator (Precision X-ray, Inc., North Branford, CT, USA) in sternal recumbency. A 40 × 40 mm square collimating cone was used to target the beam to the area of the abdomen from the diaphragm through pelvis (Figure 10), which was verified by fluoroscopy at 40 kVp and 2.5 mA with a 2 mm aluminum filter for proper alignment. Mice were irradiated with parallel-opposed fields at 90° and 270° with a single fraction of 4, 6, 8, 10, 11, or 13 Gy with a dose rate of 386 cGy/min at target depth using SSD calculations with 225 kVp and 13 mA and a 0.3 mm copper filter. Control mice (0 Gy) were anesthetized for the same amount of time but did not receive irradiation. Mice were irradiated on day 0 and then monitored and weighed daily for 7 days. On day 7, mice were euthanized and GI tissues were collected.

#### 4.3.2. Abdominal Irradiation ± Sunitinib

Fourteen-week-old mice were randomized to treatment groups of control (n = 5), sunitinib alone (n = 7), RT alone (n = 6), or RT + sunitinib (n = 7). Sunitinib suspension was delivered via oral gavage at a reported therapeutic dosage (40 mg/kg/day) [16]. On day 0, daily oral gavage (equivalent volume of vehicle control or sunitinib) was initiated and continued for 14 days. On day 7, mice were irradiated with sham RT or a single fraction of 12 Gy in the manner described above. Mice in control or sunitinib-alone groups were anesthetized for the same duration but did not receive irradiation. On day 14, mice were euthanized and GI tissues were collected.

### 4.4. Experimental Endpoints

Mice were evaluated and weighed daily. Mice were either euthanized according to IACUC-approved morbidity criteria, which included greater than 20% loss in body weight, or euthanized at a fixed endpoint of 14 days. Euthanasia was carried out via isoflurane gas anesthesia followed by cervical dislocation.

### 4.5. Tissue Collection

Small intestine (SI), cecum, and colon were harvested after euthanasia. Intestinal sections were flushed with 4% paraformaldehyde (PFA) to speed fixation and were then gently rolled. Tissues were stored in PFA for 24 h prior to being transferred to 70% ethanol to be stored until processing for histology.

### 4.6. Histopathological Evaluation and Quantitative and Semiquantitative Scoring of Injury

Formalin-fixed GI sections were embedded in paraffin, sectioned at 5 µM, and stained with hematoxylin and eosin (H&E). Prepared slides were imaged using an Olympus VS120 scanning microscope and analyzed using OlyVIA (Olympus, Waltham, MA, USA) software (version 4.1). Each section was blindly reviewed and evaluated by the first author (AR) and a veterinary pathologist (DR).

Thirty randomly selected small intestinal villi per mouse were measured for length using OlyVIA software. A measurement was taken from the mucosal/submucosal junction to the tip of each villus. Thirty colonic crypts per mouse were randomly selected and crypt area was calculated using OlyVIA software. A measurement was taken from the crypt base to the lumen for length and perpendicular to this line at the midpoint for width. The product of these two measurements was used to approximate an area for each crypt. All measurements were obtained and calculated with the evaluator blinded to the treatment group from which the tissue sample was obtained.

Semiquantitative scoring of GI injury was performed by a board-certified veterinary pathologist, again in the same blinded fashion, using previously described methods [68]. SI and colon were scored for five parameters: inflammation, crypt density/crypt loss, crypt abscesses, ulceration, and crypt hyperplasia/regeneration. Within the parameter of inflammation, a score was generated by examining the extent of separation/effacement of crypts and the presence of inflammatory infiltrates. The inflammatory cells included in this evaluation were polymorphonuclear neutrophils (PMNs), lymphocytes, plasma cells, macrophages, and giant cells (Appendix A).

### 4.7. Ki67 and CD31 Immunostaining

Formalin-fixed paraffin-embedded tissue sections were processed for immunohistochemical (IHC) staining for Ki67 to assess cell proliferation and CD31, also known as PECAM-1, to assess microvessel density (University of Colorado Research Histology Section (Aurora, CO, USA)). Four-micron thick paraffin sections were prepared for immunodetection of Ki-67 (Neomarkers/Thermo Scientific, Waltham, MA, USA; clone SP6; 1:250) and CD31 (Cell Signaling, Danvers, MA, USA; clone D8V9E; 1:100). Antigens were revealed in pH 9.5 BORG solution (Biocare Medical, Concord, CA, USA) for 10 min at 110 °C (NxGen Decloaker, Biocare) with a 10-min ambient cool down. Immunodetection of was performed on the Benchmark XT stainer (Ventana/Roche, Indianapolis, IN, USA) with primary incubations for 32 min at 37 °C (Ventana/Roche). Antibodies were detected with a modified I-VIEW detection kit (Ventana/Roche) where the secondary antibody and streptavidin HRP were replaced with Rabbit ImmPress (Vector Laboratories, Burlingame, CA, USA). The secondary antibody was replaced with full strength Rabbit ImmPress and the streptavidin-HRP was replaced with half strength Rabbit ImmPress diluted in PBS pH 7.6. Antibody-antigen complexes were visualized with diaminobenzidine from the I-VIEW kit. All sections were counterstained in Harris hematoxylin for 2 min, blued in 1% ammonium hydroxide, dehydrated in graded alcohols, cleared in xylene and coverglass mounted using synthetic resin. Negative controls to confirm the specificity of the immunostaining included omission of the primary antibody incubation step in the IHC protocol, substitution of the primary antibody diluent. Prepared slides were imaged using an Olympus VS120 scanning microscope and analyzed for the density of positive cells according to quantified count of vessels per mm^2^ using Visiopharm (Visiopharm Corporation, Wesminster, CO, USA) software (version 2023.09.3.15043). The tissues analyzed were whole SI, large intestine (LI), and cecum. Further quantification of Ki67+ cells isolated within the SI crypts was carried out via development of a crypt specific nuclei-detect application to discern between immunoreactive Ki-67 nuclei and immunonegative Ki-67 nuclei. A ratio of positive nuclei to total nuclei was calculated to provide normalized Ki-67+ percentages for inter sample comparison.

### 4.8. Statistical Analysis

Statistical analysis was performed using GraphPad Prism 9.3.1 (La Jolla, CA, USA). All data are reported as the mean ± standard error of the mean (SEM). Comparisons of effects across treatment groups were examined for significance using Fisher’s exact test (categorical data), ordinary one-way ANOVA (continuous data), or two-way ANOVA (weight comparisons between groups over time). Multiple comparisons were made using Tukey’s post hoc multiple comparisons test. Differences were considered significant for *p* ≤ 0.05.

## Figures and Tables

**Figure 1 ijms-25-01838-f001:**
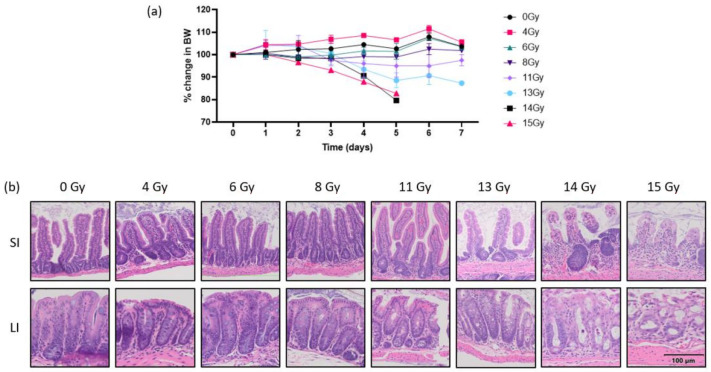
(**a**) Change in body weight relative to day 0 in mice exposed to abdominal RT. (**b**) Representative H&E-stained photomicrographs of tissue sections of SI and LI five to seven days following exposure to various doses of abdominal RT; 20× magnification. Scale bar applies to all images. RT, radiation therapy; SI, small intestine; LI, large intestine.

**Figure 2 ijms-25-01838-f002:**
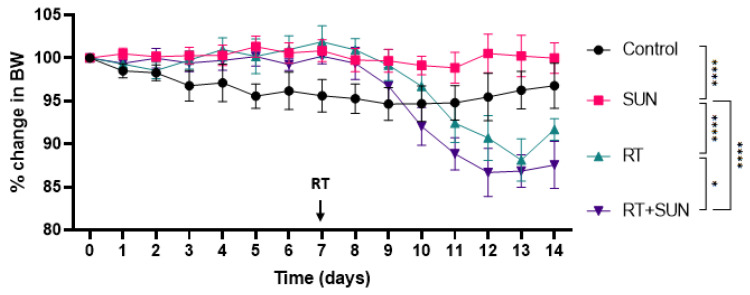
Change in body weight relative to day 0. Oral treatment initiated on day 0, and abdominal or sham RT occurred on day 7 (*p* < 0.0001) (* *p* < 0.05; **** *p* < 0.0001). RT, radiation therapy; SUN, sunitinib.

**Figure 3 ijms-25-01838-f003:**
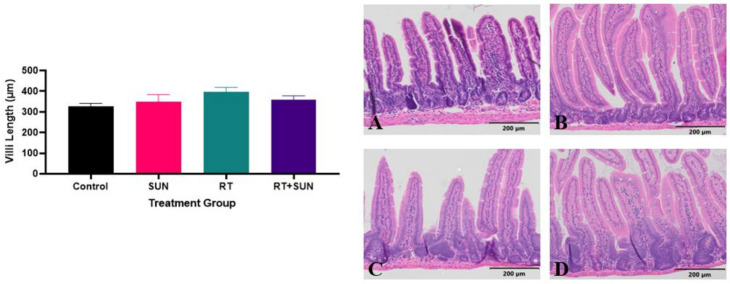
Mean (+SEM) of SI villi length for each treatment group. Photomicrographs of histological sections of colon from each of the four groups: control (**A**), sunitinib alone (**B**), RT alone (**C**), and RT + sunitinib (**D**). Magnification 10×. SI, small intestine; RT, radiation therapy.

**Figure 4 ijms-25-01838-f004:**
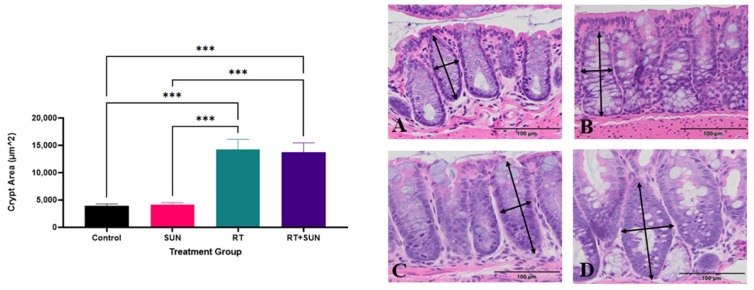
Mean (+SEM) of LI crypt area for each treatment group (*p* < 0.0001). Photomicrographs of histological sections of colon from each of the four groups: control (**A**), sunitinib alone (**B**), RT alone (**C**), and RT + sunitinib (**D**). Arrows represent examples of how area was approximated. Magnification 20× (*** *p* < 0.01). LI, large intestine; RT, radiation therapy.

**Figure 5 ijms-25-01838-f005:**
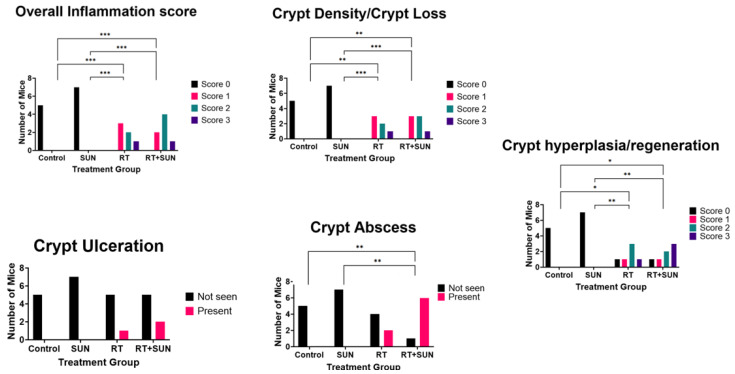
Summary of semiquantitative scoring of GI inflammation parameters (* *p* < 0.05; ** *p* < 0.01; *** *p* < 0.001). GI, gastrointestinal; SUN, sunitinib; RT, radiation therapy.

**Figure 6 ijms-25-01838-f006:**
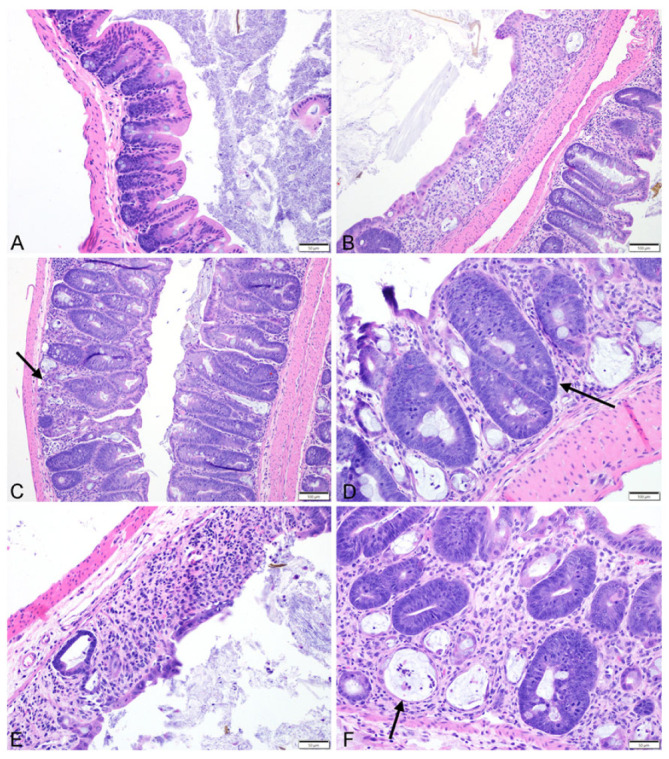
Representative photomicrographs depicting (**A**) normal mouse colon with inflammation, crypt hyperplasia, and crypt loss scores of 0; (**B**) an inflammation score of 3 and a crypt density score of 3 with segmental extensive loss of mucosal architecture characterized by complete loss of crypts, mucosal erosion and ulceration and replacement and expansion of the lamina propria by a dense inflammatory infiltrate at 10× magnification; (**C**) an inflammation score of 1 and a crypt density score of 1 with small multifocal areas of minimal disruption of the mucosal architecture characterized by ectasia and loss of few crypts (black arrow), with infiltration and mild expansion of the lamina propria by lymphocytes, plasma cells, and few neutrophils at 10× magnification; (**D**) an crypt hyperplasia score of 3 with marked crypt hyperplasia characterized by the presence of deep crypts which span the entire length of the mucosa and contain intensely basophilic epithelial cells which pile up and contain numerous mitotic figures (black arrow) at 20× magnification; (**E**) focally extensive ulceration of the mucosa, with marked crypt loss and expansion of the lamina propria by moderate numbers of infiltrating inflammatory cells at 20× magnification; (**F**) the presence of crypt microabscesses (black arrow) characterized by crypts which are moderately ectatic, lined by attenuated epithelium and contain lumina filled with few neutrophils and sloughed epithelial cell debris at 20× magnification.

**Figure 7 ijms-25-01838-f007:**
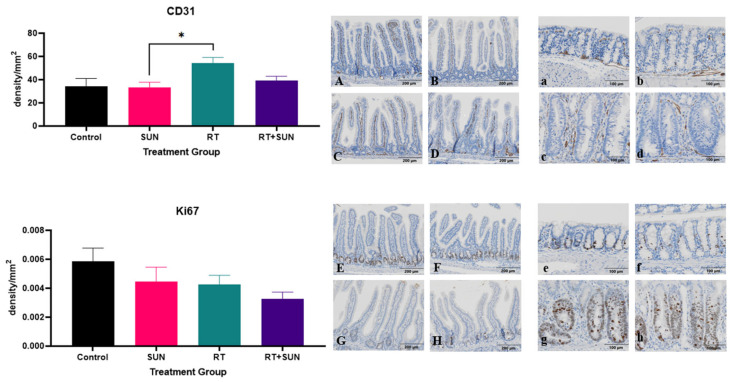
Mean (+SEM) density of positively stained cells per mm^2^ for CD31 and Ki67 for each group with representative photomicrographs. CD31: SI villi, control (**A**), sunitinib alone (**B**), RT alone (**C**), and RT + sunitinib (**D**); LI crypts, control (**a**), sunitinib alone (**b**), RT alone (**c**), and RT + sunitinib (**d**). Ki67: SI villi, control (**E**), sunitinib alone (**F**), RT alone (**G**), and RT + sunitinib (**H**); LI crypts, control (**e**), sunitinib alone (**f**), RT alone (**g**), and RT + sunitinib (**h**). SI villi magnification 10×; LI crypt magnification 20× (* *p* < 0.05). SI, small intestine; LI, large intestine; RT, radiation therapy; SUN, sunitinib.

**Figure 8 ijms-25-01838-f008:**
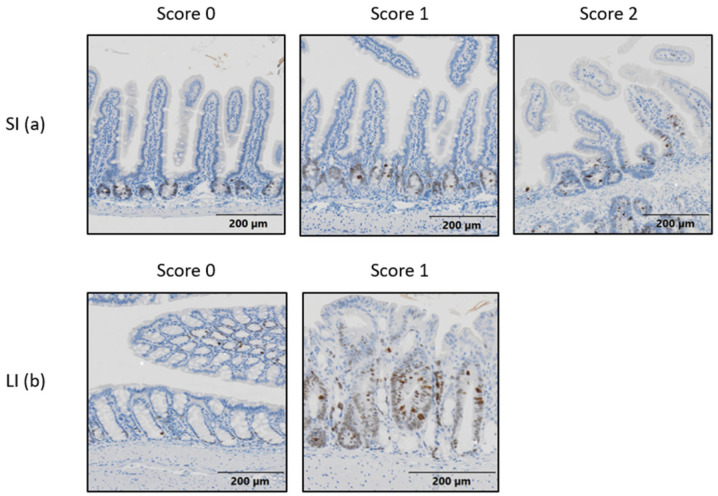
Photomicrographs showing an example of each semiquantitative score indicating Ki67+ cell distribution in SI villi (**a**) and LI crypts (**b**). Magnification 10×. SI, small intestine; LI, large intestine.

**Figure 9 ijms-25-01838-f009:**
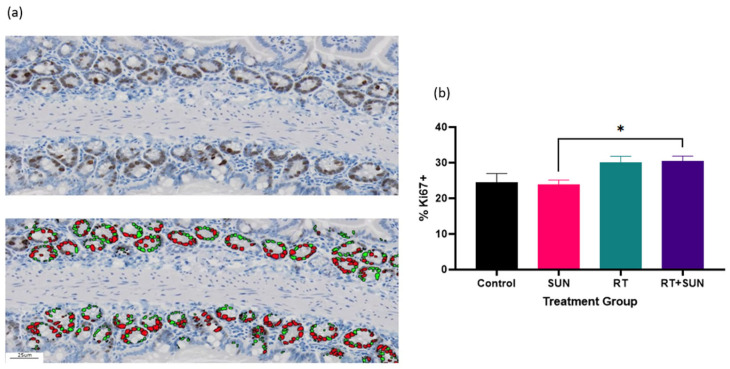
(**a**) Photomicrographs showing Ki67+ cells (brown) in the crypts of the small intestine (**top**) and the corresponding Ki67+ cells identified with VisioPharm software (version 2023.09.3.15043) (**bottom**). Note that Ki67+ nuclei are indicated by red, while Ki67 negative nuclei are green; (**b**) mean (+SEM) percentage of Ki67 positively stained cells within the crypts of the SI villi for each group (* *p* = 0.0296).

**Figure 10 ijms-25-01838-f010:**
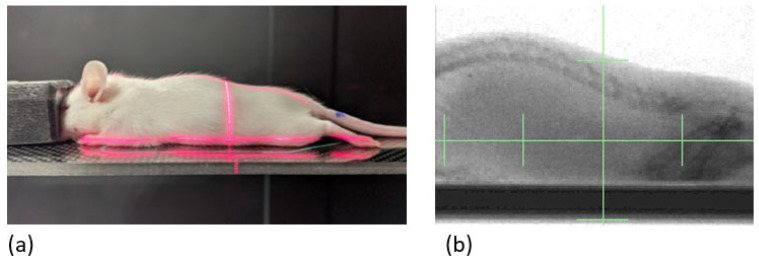
Mouse abdominal RT. (**a**) Initial setup of subject on irradiation couch; (**b**) targeted treatment field, aligned using fluoroscopy, spans from the diaphragm through the pelvis. RT, radiation therapy.

## Data Availability

The data presented in this study are available on request from the corresponding author.

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
