# Peer review of "Investigation of Gastrointestinal Toxicities Associated with Concurrent Abdominal Radiation Therapy and the Tyrosine Kinase Inhibitor Sunitinib in a Mouse Model"

_ijms, 2024, doi:10.3390/ijms25031838_

Round 1

Reviewer 1 Report

Comments and Suggestions for Authors

1.The author should further explain the novelty of this study. Line 332-Lin335 indicated that previous studies reported the post-RT GI vascularization and inflammation. In this study sunitinib seemed not to induce inflammation and authors should further elaborate their new discoveries in this study.

2. Figure 4 crypt area for each treatment group should be pointed out by arrows and further zoomed

3 Figure 5 histological figures (such as sample tissues under microscope) should be provided 

4 Figure 6 two set of bar graphs should be given  one for SI villi, another for LI crypts 

Reviewer 2 Report

Comments and Suggestions for Authors

The manuscript is devoted to the gastrointestinal toxicity of combined cancer therapy with ionizing radiation and a tyrosine kinase inhibitor (sunitinib) in a mouse model. The authors identified a lack of angiogenic response in irradiated gastrointestinal tissues when abdominal radiation treatment was combined with sunitinib, which may correlate with clinical toxicities seen in canine and human patients receiving combined treatment.

The manuscript is neatly written, with a good introduction and discussion, methods and results properly presented. My main doubt concerns the suitability of the manuscript for the journal. In my opinion it should be submitted rather to an oncologic journal, its contribution to the molecular sciences being secondary and concerning the effect of sunitinib at the organismal level in combined cancer therapy.

Minor remarks:

Figures 1,3,6, and 7. As the magnification depends on the magnification of the picture, could the Authors consider placing a dimension bar (e. g. 100 um) on the photo?

Figure 5. Smaller inscriptions are hardly visible, so some magnification of the Figure or increase the font would be advisable.

Line 498: “sectioned at 5 mM”?

Round 2

Reviewer 1 Report

Comments and Suggestions for Authors

The author has revised the manuscript properly according to reviewer's comments .